# Profiling Isokinetic Strength of Shoulder Rotator Muscles in Adolescent Asymptomatic Male Volleyball Players

**DOI:** 10.3390/sports7020049

**Published:** 2019-02-22

**Authors:** Claudio Andre Barbosa de Lira, Valentine Zimermann Vargas, Rodrigo Luiz Vancini, Marilia Santos Andrade

**Affiliations:** 1Laboratório de Avaliação do Movimento Humano, Faculdade de Educação Física e Dança, Universidade Federal de Goiás, Goiânia 74690-900, Brazil; 2Departamento de Fisiologia, Universidade Federal de São Paulo, São Paulo 04023-901, Brazil; valentinezvargas@gmail.com (V.Z.V.); marilia1707@gmail.com (M.S.A.); 3Centro de Educação Física e Desportos, Universidade Federal do Espírito Santo, Vitória 29075-910, Brazil; rodrigoluizvancini@gmail.com

**Keywords:** isokinetic dynamometer, volleyball, sport, rehabilitation, shoulder

## Abstract

The aim of the study was to describe the strength symmetry of internal and external rotator muscles and the conventional and functional strength balance ratios between these muscles in adolescent male volleyball players. Twenty-eight male adolescent volleyball players (15.5 ± 1.1 years (15–17 years); 73.2 ± 10.9 kg (55.3–100.1 kg) and 184.9 ± 8.4 cm (170–209 cm)) participated in this cross-sectional study. Concentric and eccentric peak torque of external and internal rotator muscles were measured, and conventional and functional strength balance ratios were calculated. The dominant limb presented significantly higher values for peak torque than the non-dominant limb of internal rotator muscles at concentric action assessed at 60°/s (48.7 ± 13.7 Nm and 43.9 ± 11.6 Nm, *p* = 0.01 and *d* value = 0.37) and at 240°/s (44.7 ± 11.2 Nm and 41.1 ± 11.0 Nm, *p* = 0.03 and *d* = 0.32). However, there was no difference in the peak torque of external rotator muscle between limbs for either angular speed. Regarding strength balance ratios, neither conventional (74.8 ± 14.3 for dominant limb and 80.1 ± 14.0 for non-dominant limb, *p* = 0.06 and *d* = 0.37) nor functional ratio (1.2 ± 0.4 for dominant limb and 1.3 ± 0.5 for non-dominant limb, *p* = 0.06 and *d* = 0.22) presented significant contralateral differences. Despite the short practice time, adolescent male volleyball athletes already have significant contralateral differences for internal rotator muscles and conventional ratio tends to be asymmetrical. Thus, preventive shoulder-strengthening programs, focused on the internal rotator muscles of the non-dominant limb, aiming to correct contralateral deficiency and conventional ratio, may be warranted for this population in the process of biological growth, maturation and development.

## 1. Introduction

Volleyball is characterized by repeated powerful high-speed upper-limb actions presented during training sessions and matches, which could be related to the high incidence of shoulder injuries in this sport [1,2,3,4,5]. The occurrence of these injuries is also influenced by the force of the ball impact—the ball speed can reach almost 100 km/h—as well as the high volume and intensity of training and frequent participation in matches [6]. Among the intrinsic risk factors previously associated with shoulder injuries (modified range of motion, scapulothoracic dyskinesis and muscular failure), this study focused on muscular strength. Muscular strength balance between shoulder antagonist muscles is of fundamental importance to maintain glenohumeral joint stability during upper-limb actions [7,8]. In this context, isokinetic strength testing has been extensively used to assess shoulder muscles performance [9,10,11]. Traditionally, muscle strength shoulder balance has been evaluated by the ratio between the peak torque of the external and internal rotator muscles of the glenohumeral joint, both during concentric action. This ratio has been called the conventional strength balance ratio [12,13,14]. However, this ratio does not take into account the eccentric muscular action of the antagonist muscles presented during an upper-limb action. For example, during an attack action, internal rotator muscles act concentrically in the acceleration phase, while external rotator muscles act eccentrically in the deceleration phase, which helps to slow down the movement and resist the translational movement of the humeral head [10]. The balance between these muscular actions can be evaluated by the functional strength balance ratio and is calculated by the ratio between the peak torque of the external rotator muscles during eccentric action and the peak torque internal rotator muscles during concentric action of the glenohumeral joint. Functional ratio is associated with the dynamic stability of the joint and maintaining a central position of the humeral head [8,15]. As the role of the external rotator muscles is not only to slow down the movement, but also to help to maintain the dynamic stability of the glenohumeral joint, an evaluation of the functional ratio has been considered to be more interesting, in terms of injury-prevention programs, than the conventional ratio [1,8,15,16,17,18,19,20]. Edouard et al. [20] conducted a prospective study with female handball players, which is also an overhead sport, and the authors showed that the athletes who presented muscular imbalance reported a significantly higher injury risk of 2.57 for conventional ratio and 1.50 for functional ratio. Stickley et al. [8] also concluded that shoulder dysfunction related to strength ratio imbalance may also exist in adolescent female volleyball athletes. Furthermore, another important muscular characteristic is muscular strength asymmetry (difference between right and left sides). Asymmetry strength values lower than 15% have been suggested [21,22]. Several previous studies have been developed in order to identify the muscular profile in adult volleyball players, and they usually attribute the muscular asymmetry and imbalance [5] found to the high-volume and intensity of training performed for several training years [10,23]. Although adolescent athletes do not have many years of sports experience, they are in the process of biological growth and maturation and they present incomplete neuromuscular development, which may also be a contributing factor of shoulder strength imbalance or asymmetry [8]. Although these muscular characteristics are extensive studies in adult, little is known about youth athletes [8,24]. Indeed, to the best of our knowledge, only one study has reported data on shoulder strength adaptation in youth volleyball female athletes [8], and no study has presented results from male athletes, although this information may help to guide strengthening, injury prevention and rehabilitation programs [14,25,26].

Considering the importance of muscular symmetry and muscular strength balance for joint stability and injury prevention, the aim of this study was to describe the strength symmetry of internal and external rotator muscles, as well as the conventional and functional strength balance ratio between these muscles in adolescent male volleyball players. We hypothesized that male adolescent athletes, despite their young age and, therefore, few years of training, can present these risk factors that are already described for volleyball adult athletes.

## 2. Materials and Methods

### 2.1. Participants

Twenty-eight male adolescent volleyball players participated in the study. Athletes were recruited from the Olympic Training and Research Centre (COTP), São Paulo, SP, Brazil. The COTP is a Brazilian exercise center that has youth teams of individual and collective sports that participate in local- and national-level competitions. Thus, it is reasonable to suppose that participants were highly trained. Athletes had engaged in volleyball training for a mean of 3.4 ± 1.6 years (range: 1–7 years), trained 4.4 ± 0.5 sessions/week (range: 4–5 sessions/week), and 3.8 ± 0.6 h/session (range: 2–4 h/session). After a clear explanation of the procedures, including the risks and benefits of participation, athletes and parents or legal guardians were required to sign a consent form. The physical characteristics of the group were as follows: age: 15.5 ± 1.1 years (range: 15–17 years); body mass: 73.2 ± 10.9 kg (range: 55.3–100.1 kg); and height: 184.9 ± 8.4 cm (range: 170–209 cm). The team was comprised of 11 hitters, 6 middle hitters, 2 liberos, 5 setters and 4 opposites. All athletes were asymptomatic, free from pain and presented no upper-extremity injury at the time of testing or during the year before data collection. All experimental procedures were approved by the university’s Human Research Ethics Committee (protocol number 53831115.6.0000.5505) and conformed to the principles outlined in the Declaration of Helsinki.

### 2.2. Isokinetic Strength Test

All athletes warmed up on an arm cycle ergometer (Cybex Inc., Ronkonkoma, NY, USA) for five minutes with a resistance of 25 Watts, followed by low-intensity dynamic stretching exercises for upper limbs, to avoid stretching influence in strength performance during isokinetic testing [27].

After warm-up procedures, all athletes were submitted to isokinetic muscle evaluation of the internal and external rotator muscles of the shoulders. To this end, an isokinetic dynamometer Biodex System 3 (Biodex Medical Systems, Shirley, New York, NY, USA) was used for the tests. The isokinetic dynamometer calibration procedure was performed according to the manufacturer’s recommendations. Athletes were previously informed and prepared for the evaluation according to previous studies [13,14].

Limb dominance was determined by asking the participants which limb they preferred to use for hitting or serving the ball. Both sides were assessed, and the dominant limb was assessed first.

Athletes assumed a sitting position for the muscle strength test, and standard stabilization strapping was placed around the trunk and waist to minimize additional movement and ensure the same conditions for all participants [12]. 

Internal and external rotator muscle strength was tested through 120 degrees of range of motion: between 30 degrees of internal rotation and 90 degrees of external rotation [13,14]. Athletes performed three submaximal trials to familiarize themselves with the procedure and they were tested with five repetitions for concentric action at 60 and 240°/s and five repetitions for eccentric action at 240°/s, always in this order. We chose to perform the concentric test prior to the eccentric one in order to safeguard participants, since most of them had immature musculoskeletal development, and the concentric test allows them to increase their familiarity with maximal isokinetic testing before eccentric exposure [8]. This number of repetitions has demonstrated high reliability for isokinetic testing [13,14]. A 30 s rest was given after the third submaximal trial, a one-minute break was given between two angular velocities, and a three-minute break was given when the machine setting was changed for the non-dominant side. The angular speed of 60°/s was used to evaluate the peak torque of the internal and external rotator muscle, total work and the conventional balance ratio, and 240°/s was used to evaluate the peak torque of the external rotator muscles during eccentric action, peak torque of internal rotator muscles during concentric action and the functional balance ratio. With the purpose of describing the isokinetic strength values, the peak torque and total work were also described relative to body mass (Nm/kg and J/kg, respectively) to compare athletes with different body mass. These relative peak torque and total work values were multiplied by 100. Contralateral strength deficit was calculated as: {[(peak torque of dominant side −peak torque of non-dominant side) divided by peak torque of dominant side] × 100} to assess symmetry between limbs. All athletes were tested by a single evaluator who was trained and experienced in the use of isokinetic devices. The same verbal encouragement was given to each athlete throughout the test. Visual feedback from the computer screen was not permitted. All evaluations were performed in the afternoon period to avoid the influence of the circadian cycle.

### 2.3. Statistical Analysis

The normality of the data was tested with the Shapiro−Wilk test and homogeneous variability using the Levene’s test. As all variables presented Gaussian distribution, all results of the measurements are expressed as mean and standard deviation and Student-*t* test was used for contralateral comparison. The significance level was set at *p* ≤ 0.05. Effect size calculation was adopted, besides the traditional statistical approaches. The measures of the effect size for changes in outcome were calculated by dividing the mean difference by the standard deviation of the contralateral limb measurement. Calculating Cohen’s *d* effect sizes, the magnitude of any change was judged according to the following criteria: *d* = 0.2 was considered a “small” effect size; 0.5 represented a “medium” effect size; and 0.8 a “large” effect size [28]. Confidence intervals (90%) were also calculated.

## 3. Results

Table 1 shows absolute and relative peak torque values found for concentric and eccentric activity of internal and external rotator muscle in dominant and non-dominant limbs at 60 and 240°/s, respectively. The dominant limb presented significantly higher values for concentric absolute peak torque of internal rotator muscles at 60°/s (*p* = 0.01 and *d* value = 0.37) and at 240°/s (*p* = 0.03 and *d* value = 0.32) as compared with the non-dominant limb, and the effect size of the difference was classified as medium. The same contralateral significant difference was found for relative internal rotator peak torque values, the dominant limb presenting significantly higher values at 60°/s (*p* = 0.01 and *d* value = 0.37) and at 240°/s (*p* = 0.04 and *d* value = 0.29) as compared with the non-dominant limb. On the other hand, data presented no contralateral difference for absolute and relative peak torque of external rotator muscles. Similar results were observed for total work values (Table 2). Adolescent volleyball athletes presented, in the dominant side, a higher absolute total work for internal rotator muscles values, assessed at 60°/s (*p* = 0.05 and *d* value = 0.31) and at 240°/s (*p* = 0.03 and *d* value = 0.39), as compared with the non-dominant limb, but between the sides there was no difference of total work for the external rotator. When the peak torque of the internal rotator muscles values were presented relativized by body mass, the same contralateral difference can be seen. The dominant side presented significantly higher values for relative total work for internal rotators assessed at 60°/s (*p* = 0.04 and *d* value =0.31) and at 240°/s (*p* = 0.03 and *d* value = 0.38) as compared with the non-dominant limb

Table 3 shows values for conventional and functional balance ratios for the dominant and non-dominant upper limbs at 60 and 240°/s; no contralateral differences were found in these variables.

## 4. Discussion

The aim of this study was to verify the existence of asymmetric muscular strength of external and internal rotator muscles of the shoulder and to evaluate the muscular strength balance between them in adolescent male asymptomatic volleyball players. Despite the short time of volleyball practice, our hypothesis was that already in adolescent these risk factors for shoulder injuries (muscular asymmetry or imbalance) may be present. The main result of this study was that adolescent athletes present asymmetrical internal rotator strength values (dominant limb is stronger), but have no difference in external rotator values. This result produces a muscle imbalance trend in the dominant side, which, while not being significant (*p* = 0.06), had a positive effect size (*d* = 0.37, CI = 0.05 to 0.69). With regards to muscular strength balance, an individual analysis showed that 14% of the athletes presented balance ratio on the dominant side lower than the literature recommended values (66–75%) [9], while 39% of the athletes presented balance ratio higher than the recommended values.

One known risk factor for shoulder injuries is strength contralateral differences higher than 15% [21,29,30,31]. Trakis et al. [22] showed that muscular strength asymmetry was associated with the presence of shoulder joint pain. The presence of this strength asymmetry has been previously shown in athletes who are involved in asymmetric sports [10,13,14]. Probably, this bilateral difference results from a high volume and intensity of sports training. In the present study, volleyball athletes presented significant contralateral strength difference (in peak torque and total work values) for internal rotator shoulder muscles. However, it is important to note that the effect size of all internal rotators muscles contralateral differences was classified as medium (0.20 < *d* < 0.50), and the mean of bilateral difference of the peak torque of internal rotators muscles was 7.3 ± 18.1%, which is, on average, lower than the recommended value. In fact, some muscular strength asymmetry should be expected for asymmetric overhead sport, and, to a certain degree, it may not be detrimental to volleyball performance or to the shoulder joint. However, it is also important to perform an individual analysis of the data. Fourteen athletes for the present study had contralateral difference greater than 15%. In other words, 50% of volunteers presented higher asymmetric strength values than the recommendation. As the mean values of the asymmetric strength was lower than 15%, a non-dominant-limb strengthening program for the whole group is not recommended. On the other hand, considering that half of the sample presented internal rotator muscle strength asymmetry greater than the recommendation, an individual internal rotator muscle strength evaluation is suggested, aimed at identifying those who present higher values of strength asymmetry, which may compromise the health of the shoulder joint and, for these, a specific strengthening for the non-dominant side is recommended to improve contralateral deficiency. 

It is not possible to compare current data to those published by Stickley et al. [8], who also studied adolescent volleyball athletes, but female, because they presented only the dominant upper-limb strength data. Nevertheless, the underdeveloped muscles presented in immature subjects, associated with asymmetrical characteristics of volleyball, may contribute to these asymmetrical strength forces and to the appearances of injury risk factors in adolescence [32]. 

Another considered risk factor for shoulder injury is the presence of muscular strength imbalance between internal and external rotators muscles. Codine et al. [33] conducted a review study and showed that the relationship between agonist and antagonist muscles provides important information on the pathological condition of the joint. The same authors also showed that, in cases of impingement syndrome and glenohumeral instability, the muscular strength balance is impaired, and this change seems to be related more to the cause than to the consequence of these injuries. In a prospective study, Edouard et al. [20] showed that the relative risk for shoulder injuries was 2.57 times higher (95% CI: 1.60–3.54; *p* < 0.05) if handball players had an imbalanced muscular strength profile. In this context, conventional ratio (measured at 60°/s) should be between 66% and 75% as recommended by Ellenbecker and Davies [9].

Comparing our results with the literature recommended values, we observed that the mean value for the non-dominant side (80 ± 14%) was higher than that recommended in the literature, suggesting a higher risk of injury. Interestingly, female volleyball players with a history of shoulder injuries presented a similar mean of conventional ratio (79%), while those who had not experienced previous injuries produced lower values for conventional ratio (75%) [8], which is accordance with a hypothesis of the existence of a maximal value for conventional ratio (66–75%) [9] that is ideal to prevent injuries. On the other hand, previous data for adult volleyball players showed lower values for conventional ratio. Hadzic et al. [10] showed 61% for the dominant limb and 63% for the non-dominant limb, and Gozlan et al. [23] presented 50% and 52% for the dominant and non-dominant limbs, respectively. It is possible that the years of training will cause a decrease in the conventional ratio, possibly due to the greater strength gain of the internal rotator muscles than in the external rotators muscles, as was demonstrated for handball players [34]. 

Analyzing the bilateral conventional ratio difference at 60°/s, we observed that, despite having no significant contralateral difference (*p* = 0.06), the effect size was classified as medium (*d* = 0.37), which creates the need for attention to be paid to this probable asymmetry of strength balance. It is possible that young volleyball athletes would benefit from non-dominant internal rotator muscles strengthening to reduce strength asymmetry and to improve conventional ratio. 

Nowadays, greater emphasis has been placed on the eccentric strength of antagonist muscles, because of their importance for dynamic joint stabilization [1,17]. Stability of the glenohumeral joint during acceleration, deceleration, and the follow-through phases of striking is provided by the rotator cuff muscles acting eccentrically to compress the humeral head. Active and passive mechanisms maintain dynamic stabilization and compression of the humeral head in the glenoid fossa during spiking and serving [8]. As the upper extremity accelerates through its range of motion, the supraspinatus, infraspinatus, and teres minor muscles eccentrically resist translation of the humeral head and assist in the deceleration of the moving limb. Consequently, rotator cuff weakness allows increased stress to be placed on the passive stabilizers of the shoulder, leading to detrimental translation of the humeral head [8]. For this reason, functional ratio has been widely studied recently [1,13,20,34]. Value higher than 1.0 for functional ratio have been suggested for dynamic shoulder joint stability [1]. Functional ratio values above 1.0 indicate that the strength of the eccentric external rotator muscles is greater than the strength of concentric internal rotator muscles. This greater strength may be needed to slow the fast-moving upper limb [13]. Adult volleyball players showed a functional ratio higher than 1.0 [12], and those who present a shoulder injury history present a lower functional ratio than those without injury [8]. In the present study, adolescent volleyball players presented functional ratio values higher than 1.0 for both upper limbs.

To the best of our knowledge, this is the first study aimed at verifying the presence of strength asymmetry and imbalance between shoulder joint muscles in male adolescents’ volleyball players. 

## 5. Conclusions

Despite the short practice time, half of adolescent male volleyball athletes already had contralateral difference for internal rotator muscles higher than 15% and conventional rotators tended to be asymmetrical. Moreover, the mean value for conventional ratio in the non-dominant limb was higher than the recommended literature value. Thus, preventive shoulder strengthening programs, focused on non-dominant limb internal rotator muscles and aiming to correct contralateral deficiency and conventional ratio may be warranted even for adolescent volleyball athletes.

## Figures and Tables

**Table 1 sports-07-00049-t001:** Absolute and relative isokinetic peak torque (PT) during concentric (con) and eccentric (ecc) actions, for dominant (D) and non-dominant (ND) upper limbs, assessed in 60 and 240°/s at internal (IR) and external (ER) shoulder rotations.

Velocity °/s (Action Type)	D	ND	Contralateral Difference (%)	*p* Value	*d* Value (90% CI)
**60 (con)**					
Absolute PT of IR (Nm)	48.7 ± 13.7	43.9 ± 11.6 *	7.3 ± 18.1	0.01	0.37 (0.14 to 0.60)
Relative PT of IR (Nm/kg)	66.8 ± 17.7	60.5 ± 15.9 *	7.3 ± 18.1	0.01	0.37 (0.14 to 0.60)
Absolute PT of ER (Nm)	35.8 ± 11.0	35.0 ± 10.7	1.4 ± 13.3	0.40	0.07 (−0.07 to 0.21)
Relative PT of ER (Nm/kg)	48.6 ± 11.8	47.7 ± 12.5	1.4 ± 13.1	0.43	0.07 (−0.08 to 0.22)
**240 (con)**					
Absolute PT of IR (Nm)	44.7 ± 11.2	41.1 ± 11.0 *	7.1 ± 17.7	0.03	0.32 (0.08 to 0.56)
Relative PT of IR (Nm/kg)	61.4 ± 14.3	56.9 ± 15.8 *	7.1 ± 17.7	0.04	0.29 (0.06 to 0.52)
Absolute PT of ER (Nm)	31.7 ± 9.1	30.7 ± 8.2	1.7 ± 13.9	0.26	0.11(−0.05 to 0.27)
Relative PT of ER (Nm/kg)	43.2 ± 10.3	41.9 ± 9.1	1.7 ± 13.9	0.25	0.13 (−0.05 to 0.32)
**240 (ecc)**					
Absolute PT of IR (Nm)	66.5 ± 24.3	64.2 ± 21.3	1.2 ± 16.4	0.32	0.10 (−0.07 to 0.27)
Absolute PT of ER (Nm)	59.2 ± 16.3	57.7 ± 14.3	0.2 ± 16.5	0.44	0.09 (−0.11 to 0.29)

Data are mean ± SD. CI, confidence interval. * *p* < 0.05 (D vs. ND).

**Table 2 sports-07-00049-t002:** Total work (TW) of internal (IR) and external (ER) rotations for dominant (D) and non-dominant (ND) upper limbs at 60 and 240°/s.

Velocity (°/s)	D	ND	Contralateral Difference (%)	*p* Value	*d* Value (90% CI)
**60**					
Absolute TW of IR (J)	59.0 ± 15.6	54.0 ± 16.0	6.4 ± 21.8	0.05	0.31 (0.05 to 0.37)
Relative TW of IR (J/kg)	81.2 ± 20.5	74.5 ± 22.1 *	6.4 ± 21.8	0.04	0.31 (0.06 to 0.55)
Absolute TW of ER (J)	43.0 ± 13.0	41.8 ± 12.9	1.6 ± 17.2	0.42	0.09 (−0.10 to 0.28)
Relative TW of ER (J/kg)	58.4 ± 13.9	57.1 ± 15.7	1.6 ± 17.2	0.13	0.08 (−0.02 to 0.59)
**240**					
Absolute TW of IR (J)	51.4 ± 15.2	45.7 ± 13.9 *	8.5 ± 23.2	0.03	0.39 (0.10 to 0.68)
Relative TW of IR (J/kg)	70.8 ± 20.0	63.3 ± 19.6 *	8.5 ± 23.2	0.03	0.38 (0.10 to 0.66)
Absolute TW of ER (J)	35.1 ± 11.1	33.4 ± 9.9	2.8 ± 18.6	0.14	0.16 (−0.02 to 0.34)
Relative TW of ER (J/kg)	47.9 ± 12.6	45.6 ± 11.6	2.8 ± 18.6	0.14	0.19 (−0.02 to 0.40)

Data are mean ± SD. CI, confidence interval. * *p* < 0.05 (D vs. ND).

**Table 3 sports-07-00049-t003:** Conventional (CR) and functional (FR) strength balance ratio for the dominant (D) and non-dominant (ND) upper limbs at 60 and 240°/s angular speeds.

Velocity (°/s)	D	ND	*p* Values	*d* Value (90% CI)
**60**				
CR (%)	74.8 ± 14.3	80.1 ± 14.0	0.06	0.37 (0.05 to 0.69)
**240**				
CR (%)	71.4 ± 12.6	77.3 ± 19.2	0.14	0.36 (−0.04 to 0.76)
FR (%)	1.2 ± 0.4	1.3 ± 0.5	0.06	0.22 (0.02 to 0.41)

Data are mean ± SD. CI, confidence interval.

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
