# Peer review of "Profiling Isokinetic Strength of Shoulder Rotator Muscles in Adolescent Asymptomatic Male Volleyball Players"

_sports, 2019, doi:10.3390/sports7020049_

Reviewer 1 Report

Please have this manuscript carefully checked for English grammar.

Make sure all the appropriate italicizing is done, for example the t in t-test, the p in p-value, the d in Cohen’s d, etc.

Abstract

Could the authors kindly provide the mean +/- SD or CI for peak torque values in the abstract?

Line 14: This first statement is not a complete sentence, please revise

Line 19: “dominant limb presented significantly higher values for peak torque…” higher than what?

Introduction

The first paragraph does not clearly draw the line between the information and the research question.

More context is needed from the studies demonstrating the importance of muscular balance in prevention of these injuries. Please consider adding some of the specifics of these studies.

Line 60: The studies referenced for the dangers of right-left asymmetry are all in the lower extremity (except for 1 study which was a review) and performed on athletes of bilateral sports or actions (e.g. sprinting). This does not have the same meaning in sports or actions where there is a unilateral action (e.g. hitting motion in volleyball). In fact, the asymmetries noted in these unilateral-type athletes are not often a detriment to their performance. It is a common misconception that athletes need to be in perfect balance when their sport inherently necessitates that they are not. That being said, I agreed with the assessment in the abstract which outlined that in youth athletes it is of more concern because they are still developing and likely playing multiple sports. I would like to see the focus being on proper neuromuscular development in these young athletes rather than preventing a future imbalance in their sport.

Methods

Lines 136-144: Please indicate which effect size is being used. It seems that it is the Cohen’s d effect size but please put this in the text.

Results

There is a sample table in here (noted as “Table 1”) that is from the guidelines provided by Sports. Please remove this sample table.

Table 1- In the footnotes, please define all terms even if they are defined in the main text (D= dominant, for example)

Discussion

The discussion often suffers from a lack of practical application. While these data certainly have practical application, the authors would be well-served to discuss it more thoroughly. Currently, the discussion describes the mechanism of the results and their relationship to previous findings, but don’t take the next step of making it applicable for athletes and coaches.

Line 194: The authors bring up the previously noted asymmetries in unilateral/asymmetric sport actions (references 12-14), but do not discuss this further. It would be useful to the readers to discuss the pros and cons of this. In reality, the issue of asymmetry in these athletes probably lies somewhere in the middle (at least in adults)- we shouldn’t overreact to asymmetries such that we are focused solely on creating balance and therefore impacting performance training, but we should also pay mind to these asymmetries in order to avoid extreme imbalances that could affect injury and overall health. A discussion here is warranted.

Author Response

Manuscript ID: sports-407806

Title: Profiling isokinetic strength of shoulder rotator muscles in

adolescent asymptomatic male volleyball players

30-Jan-2019

Prof. Dr. Eling Douwe de Bruin

Editor-in-Chief

Sports

 Dear Editor,

            We would like to thank the reviewers for their thorough review and insightful feedback; we have made all necessary revisions (highlighted in the text with MS Word track changes) and answered the reviewers’ questions point by point. The manuscript has been improved substantially and we hope it is now suitable for publication in Sports.

Reviewer #1

Comments and Suggestions for Authors

Please have this manuscript carefully checked for English grammar.

Answer: The manuscript was reviewed by an English native speaker (proofread by Proof-Reading-Service.com).

Make sure all the appropriate italicizing is done, for example the t in t-test, the p in p-value, the d in Cohen’s d, etc.

Answer: Thank you for calling this to our attention. We have corrected this error.

Abstract

Could the authors kindly provide the mean +/- SD or CI for peak torque values in the abstract?

Answer: The mean +/- SD values have been included as requested by you.

Line 14: This first statement is not a complete sentence, please revise

Answer: Thank you for calling this to our attention. The sentence has been rewritten.

Line 19: “dominant limb presented significantly higher values for peak torque…” higher than what?

Answer: Higher than non-dominant limb. We have reworded the sentence to clarify and meet with your expectation. Thank you for your constructive comment.

Introduction

The first paragraph does not clearly draw the line between the information and the research question.

Answer: The first paragraph has been rewritten to better draw the line between the information and the research question.

More context is needed from the studies demonstrating the importance of muscular balance in prevention of these injuries. Please consider adding some of the specifics of these studies.

Answer: We have included previous studies that demonstrated the relevance of strength imbalance as a risk factor for shoulder injuries. Please let us know if this explanation does not resolve your doubts in this matter.

Line 60: The studies referenced for the dangers of right-left asymmetry are all in the lower extremity (except for 1 study which was a review) and performed on athletes of bilateral sports or actions (e.g. sprinting). This does not have the same meaning in sports or actions where there is a unilateral action (e.g. hitting motion in volleyball). In fact, the asymmetries noted in these unilateral-type athletes are not often a detriment to their performance. It is a common misconception that athletes need to be in perfect balance when their sport inherently necessitates that they are not. That being said, I agreed with the assessment in the abstract which outlined that in youth athletes it is of more concern because they are still developing and likely playing multiple sports. I would like to see the focus being on proper neuromuscular development in these young athletes rather than preventing a future imbalance in their sport.

Answer: We have made extensive changes in manuscript to meet your important considerations.  Please let us know if this explanation does not resolve your doubts in this matter.

Methods

Lines 136-144: Please indicate which effect size is being used. It seems that it is the Cohen’s d effect size but please put this in the text.

Answer: We used Cohen’s d effect size. We habe included this information in the manuscript to clarify this point.

Results

There is a sample table in here (noted as “Table 1”) that is from the guidelines provided by Sports. Please remove this sample table.

Answer: Thank you for calling this to our attention. We have deleted the sample table.

Table 1- In the footnotes, please define all terms even if they are defined in the main text (D= dominant, for example)

Answer: Thank you for your comment. We have defined the terms in the tables’ titles, therefore, we do not think that it is necessary to define them again in the footnotes. Please let us know if this explanation does not resolve your concerns in this matter.

Discussion

The discussion often suffers from a lack of practical application. While these data certainly have practical application, the authors would be well-served to discuss it more thoroughly. Currently, the discussion describes the mechanism of the results and their relationship to previous findings, but don’t take the next step of making it applicable for athletes and coaches.

Answer: The practical application has been included in the discussion section as requested by you. Thank you for your constructive comment.

Line 194: The authors bring up the previously noted asymmetries in unilateral/asymmetric sport actions (references 12-14), but do not discuss this further. It would be useful to the readers to discuss the pros and cons of this. In reality, the issue of asymmetry in these athletes probably lies somewhere in the middle (at least in adults)- we shouldn’t overreact to asymmetries such that we are focused solely on creating balance and therefore impacting performance training, but we should also pay mind to these asymmetries in order to avoid extreme imbalances that could affect injury and overall health. A discussion here is warranted.

Answer: A discussion about the importance of asymmetrical strength values has been included as requested by you. Please let us know if this explanation does not resolve your doubts in this matter.

Reviewer 2 Report

The article titled “Profiling isokinetic strength of shoulder rotator muscles in adolescent asymptomatic male volleyball players” it’s a very interesting paper.As you affirm in literature there aren’t information about the presence of strength asymmetry and imbalance between shoulder joint muscles of adolescent asymptomatic male volleyball players, so it’s original.Despite the number of subjects recruited it’s low, only 28, and the short practice time, many variables have been considered and interesting results were obtained. I would clarify the tables in order to make them more intuitive.

References are very good listed, clear and not too dated 

Only few minor points before publication:

line 39: repetition of the word 'thoracic'

line 40: write author’s name of the papers 

line 84: suppose-that, not supposethat

line 106: write author’s name of the papers

line 125: to avoid the repetition of “in order to”(line 124-125-126), you could use a synonymous as “with the purpose of”

line 126: …with literature data. Add references, because is not too clear what you mean 

Table 1: highlighted in bold 240 (ecc)

Table 2: what is BW, add to line 174

Table 3 is smaller than the other two

line 184: maybe after “muscles of” is missing “the shoulder”

line 185: adolescent male, not adolescentmale 

line 195, 205, 211, 216, 218, 225, 226: write author’s name of the papers, not only the number of the reference

line 346: year of reference articles need to be highlighted in bold

Author Response

Manuscript ID: sports-407806

Title: Profiling isokinetic strength of shoulder rotator muscles in

adolescent asymptomatic male volleyball players

30-Jan-2019

Prof. Dr. Eling Douwe de Bruin

Editor-in-Chief

Sports

Dear Editor,

            We would like to thank the reviewers for their thorough review and insightful feedback; we have made all necessary revisions (highlighted in the text with MS Word track changes) and answered the reviewers’ questions point by point. The manuscript has been improved substantially and we hope it is now suitable for publication in Sports.

Reviewer #2

Comments and Suggestions for Authors

The article titled “Profiling isokinetic strength of shoulder rotator muscles in adolescent asymptomatic male volleyball players” it’s a very interesting paper.As you affirm in literature there aren’t information about the presence of strength asymmetry and imbalance between shoulder joint muscles of adolescent asymptomatic male volleyball players, so it’s original.Despite the number of subjects recruited it’s low, only 28, and the short practice time, many variables have been considered and interesting results were obtained. I would clarify the tables in order to make them more intuitive.

References are very good listed, clear and not too dated 

Answer: Thank you for your constructive comment. We have changed the tables as requested by you.

line 39: repetition of the word 'thoracic'

Answer: Thank you for calling this to our attention. We have deleted one ‘thoracic’.

line 40: write author’s name of the papers 

Answer: Thank you for calling this to our attention. We have included the authors’ names.

line 84: suppose-that, not supposethat

Answer: Thank you for calling this to our attention. We have corrected this error.

line 106: write author’s name of the papers

Answer: We have reworded the sentence to clarify and meet with your expectations.

line 125: to avoid the repetition of “in order to”(line 124-125-126), you could use a synonymous as “with the purpose of”

Answer: We have reworded the sentence as requested by you.

line 126: …with literature data. Add references, because is not too clear what you mean 

Answer: We have reworded the sentence in order to clarify the meaning here.

Table 1: highlighted in bold 240 (ecc)

Answer: We have highlighted in bold ‘240 (ecc)’ as requested by you.

Table 2: what is BW, add to line 174

Answer: BW means body weight. As indicated by you in your initial comment, we have changed the tables to become its more intuitive.

Table 3 is smaller than the other two

Answer: Table 3 is smaller than the other two, because Table 3 has five columns and the other tables have six columns. Please let us know if this explanation does not resolve your concerns in this matter.

line 184: maybe after “muscles of” is missing “the shoulder”

Answer: Thank you for calling this to our attention.

line 185: adolescent male, not adolescentmale 

Answer: Thank you for calling this to our attention.

line 195, 205, 211, 216, 218, 225, 226: write author’s name of the papers, not only the number of the reference

Answer: We have included the authors’ names.

line 346: year of reference articles need to be highlighted in bold

Answer: We have highlighted the year, as requested by you.

Round  2

Reviewer 1 Report

I commend the authors for making the necessary changes to the manuscript. I believe it has been greatly improved and merits publication in the journal. Good work!

Reviewer 2 Report

Accepted in present form